# A recurrent cancer-associated substitution in DNA polymerase ε produces a hyperactive enzyme

Xuanxuan Xing[1,4], Daniel P. Kane[1,5], Chelsea R. Bulock[1], Elizabeth A. Moore[1], Sushma Sharma[2], Andrei Chabes[2,3] & Polina V. Shcherbakova[1]

Alterations in the exonuclease domain of DNA polymerase ε (Polε) cause ultramutated tumors. Severe mutator effects of the most common variant, Polε-P286R, modeled in yeast suggested that its pathogenicity involves yet unknown mechanisms beyond simple proof-reading deficiency. We show that, despite producing a catastrophic amount of replication errors in vivo, the yeast Polε-P286R analog retains partial exonuclease activity and is more accurate than exonuclease-dead Polε. The major consequence of the arginine substitution is a dramatically increased DNA polymerase activity. This is manifested as a superior ability to copy synthetic and natural templates, extend mismatched primer termini, and bypass secondary DNA structures. We discuss a model wherein the cancer-associated substitution limits access of the 3'-terminus to the exonuclease site and promotes binding at the polymerase site, thus stimulating polymerization. We propose that the ultramutator effect results from increased polymerase activity amplifying the contribution of Polε errors to the genomic mutation rate.

---

[1] Eppley Institute for Research in Cancer and Allied Diseases, Fred & Pamela Buffett Cancer Center, University of Nebraska Medical Center, Omaha, NE 68198, USA. [2] Department of Medical Biochemistry and Biophysics, Umeå University, 901 87 Umeå, Sweden. [3] Laboratory for Molecular Infection Medicine Sweden, Umeå University, 901 87 Umeå, Sweden. [4] Present address: Comprehensive Cancer Center, Ohio State University, Columbus, OH 43210, USA. [5] Present address: Department of Biological and Environmental Sciences, Le Moyne College, Syracuse, NY 13214, USA. Correspondence and requests for materials should be addressed to P.V.S. (email: pshcherb@unmc.edu)

The fidelity of DNA replication is contingent upon the serial action of DNA polymerase selectivity, exonucleolytic proofreading, and DNA mismatch repair (MMR)[1–3]. In eukaryotes, three replicative DNA polymerases, Polα, Polδ, and Polε, contribute to genome stability via their intrinsic nucleotide selectivity. Polδ and Polε are additionally equipped with a proofreading exonuclease activity that can remove incorrectly inserted nucleotides from the primer terminus, further improving the fidelity of DNA synthesis. Somatic alterations in the exonuclease domain of Polε are commonly found in hypermutated colorectal and endometrial tumors, and, at a lower frequency, in other types of gastrointestinal and gynecological cancers, as well as tumors of the brain, breast, prostate, lung, kidney, and bone[4–8]. Germline mutations affecting the exonuclease domain of Polε cause a colorectal cancer predisposition syndrome characterized by early disease onset and multiple tumors[9]. Polε-mutant tumors typically have an exceptionally high mutation load and are classified as ultramutated to distinguish them from less severely hypermutated MMR-deficient tumors.

It was originally suggested that the changes in Polε promote ultramutation by disabling proofreading[9]. Many of the cancer-associated amino acid substitutions were predicted by in silico analysis to affect DNA binding in the exonuclease site and/or exonuclease activity. Indeed, cancer-associated variants were shown to reduce exonuclease activity and fidelity of a purified catalytic fragment of human Polε[10]. However, several observations are difficult to reconcile with the idea that the pathogenicity of Polε variants results solely from adverse effects on proofreading. First, mutations at catalytic residues in the exonuclease domain, which are well known to inactivate proofreading, are rarely or never seen in tumors. Instead, mutations at other conserved residues appear as recurrent hotspots, the P286R substitution being by far the most common in sporadic cancers[4]. Second, modeling of the P286R variant in yeast produced an exceptionally strong mutator phenotype exceeding that of an exonuclease-dead Polε mutant by two orders of magnitude[11]. Mirroring these observations, $Pole^{P286R}$ mice are dramatically more cancer-prone than $Pole$ exonuclease-deficient mice[12]. The mutator effects of many other, less common, Polε variants also exceed the effects of exonuclease deficiency[13]. These observations strongly argue that the development of an ultramutated phenotype requires some functional changes in the protein distinct from a loss of proofreading. The nature of these changes and the mechanism through which the cancer-associated Polε variants elevate genome instability remain enigmatic.

In this work, we purified the yeast analog of Polε-P286R (yPolε-P301R) as a four-subunit holoenzyme and demonstrated that it is not less accurate than proofreading-deficient Polε (exo⁻ Polε). In fact, Polε-P301R is slightly more accurate, in line with the presence of residual exonuclease activity. At the same time, the analysis of mutational specificity and synergistic interactions with a MMR defect argues that the ultramutator effect in vivo results from a catastrophically high rate of errors made by Polε-P301R during replicative DNA synthesis. We found that the major property distinguishing Polε-P301R from both the wild-type and exo⁻ Polε is an extremely robust DNA polymerase activity. This is evident from a more efficient overall DNA synthesis and also a greatly improved ability to handle a variety of difficult DNA substrates that normally present an obstacle for Polε. Taking into account the structural insights provided by the companion study by Parkash and co-authors[14], we propose that the uniquely strong pathogenic effects of this recurrent cancer-associated variant result from the arginine side chain restricting access of the primer terminus to the exonuclease active site. The inability to position the 3′-terminus in the exonuclease site makes Polε a more efficient DNA polymerase, a consequence that is not achieved by simple elimination of catalytic residues. These findings provide insight into the molecular mechanisms that drive the development of ultramutated cancers, and also have implications for understanding the normal physiological role of Polε in DNA replication and mutation avoidance.

## Results

**Polε-P301R is more accurate than exonuclease-deficient Polε.** The mutator effect of the yeast Polε-P301R mimicking human Polε-P286R greatly exceeds that of any previously studied Polε mutation[11], suggesting that the enzyme might possess some unusual novel properties. A decrease in 3′→5′ exonuclease activity was expected from previous studies[10] but would be insufficient to explain the strong mutator phenotype. We first hypothesized that the P301R substitution resulted in a more severe reduction in the enzyme's fidelity, perhaps due to a combination of the impaired proofreading with a nucleotide selectivity defect. We purified the four-subunit Polε-P301R and compared its exonuclease activity and the overall fidelity to those of the wild-type Polε and exo⁻ Polε. The exo⁻ Polε is completely devoid of exonuclease activity due to the replacement of the catalytic residues Asp290 and Glu292 with alanines[15]. Polε-P301R was readily purified as a four-subunit holoenzyme with the proper stoichiometry, indicating that the mutation does not affect interaction with the accessory subunits (Supplementary Fig. 1). The four-subunit Polε-P301R showed a significantly reduced but still detectable 3′→5′ exonuclease activity in assays with a correctly matched oligonucleotide primer/template substrate (Fig. 1a), similar to previous observations with the catalytic fragment of human Polε-P286R[10]. The exonuclease activity was mildly stimulated by the presence of a mismatched base pair at, or in the vicinity of, the primer terminus (Supplementary Fig. 2) and was the highest with a single-stranded oligonucleotide substrate (Fig. 1b). Thus, Polε-P301R was clearly capable of hydrolyzing 3′-termini, although it was severely impaired in comparison to the wild-type enzyme.

Next, we characterized the fidelity of DNA synthesis by Polε-P301R in vitro using the M13mp2 lacZ forward mutation assay[16]. We previously observed that mimicking the physiological intracellular dNTP concentrations in the in vitro fidelity assays can be critical to recapitulate the mutator properties of replicative DNA polymerase variants[17]. We found that the sizes of dNTP pools in the wild-type strain and the yeast pol2-P301R mutant producing Polε-P301R were similar (Supplementary Fig. 3). The pol2-4 strains producing exo⁻ Polε are also known to have wild-type dNTP levels[18]. Accordingly, we used dNTP concentrations calculated for wild-type S-phase yeast cells[17] (see Methods) in the in vitro fidelity assay to mimic the intracellular conditions. We observed that Polε-P301R was more accurate than exo⁻ Polε. The lacZ mutant frequencies were 0.012 and 0.032 for the two enzymes, respectively ($p < 0.00001$, Fisher's exact test; Fig. 1c and Supplementary Table 1). The lower error rate of Polε-P301R was in agreement with the presence of a limited exonuclease activity (Fig. 1a, b) but in striking contrast to its prodigiously higher mutator effect in vivo[11]. Error rates were lower in Polε-P301R reactions for all types of base-base mispairs in comparison to exo⁻ Polε reactions, but a particularly strong difference was seen for transversion-type (pyrimidine-pyrimidine and purine-purine) mispairs (Fig. 1d; Supplementary Table 1), possibly because these are proofread more efficiently by the weak exonuclease of Polε-P301R. Notably, Polε-P301R-induced base substitutions occurred at a smaller number of sites (Supplementary Fig. 4a,b), suggesting that Polε-P301R is rather accurate at most DNA sequences, and there are only certain positions where its fidelity is compromised. Overall, the in vitro assays showed that Polε-

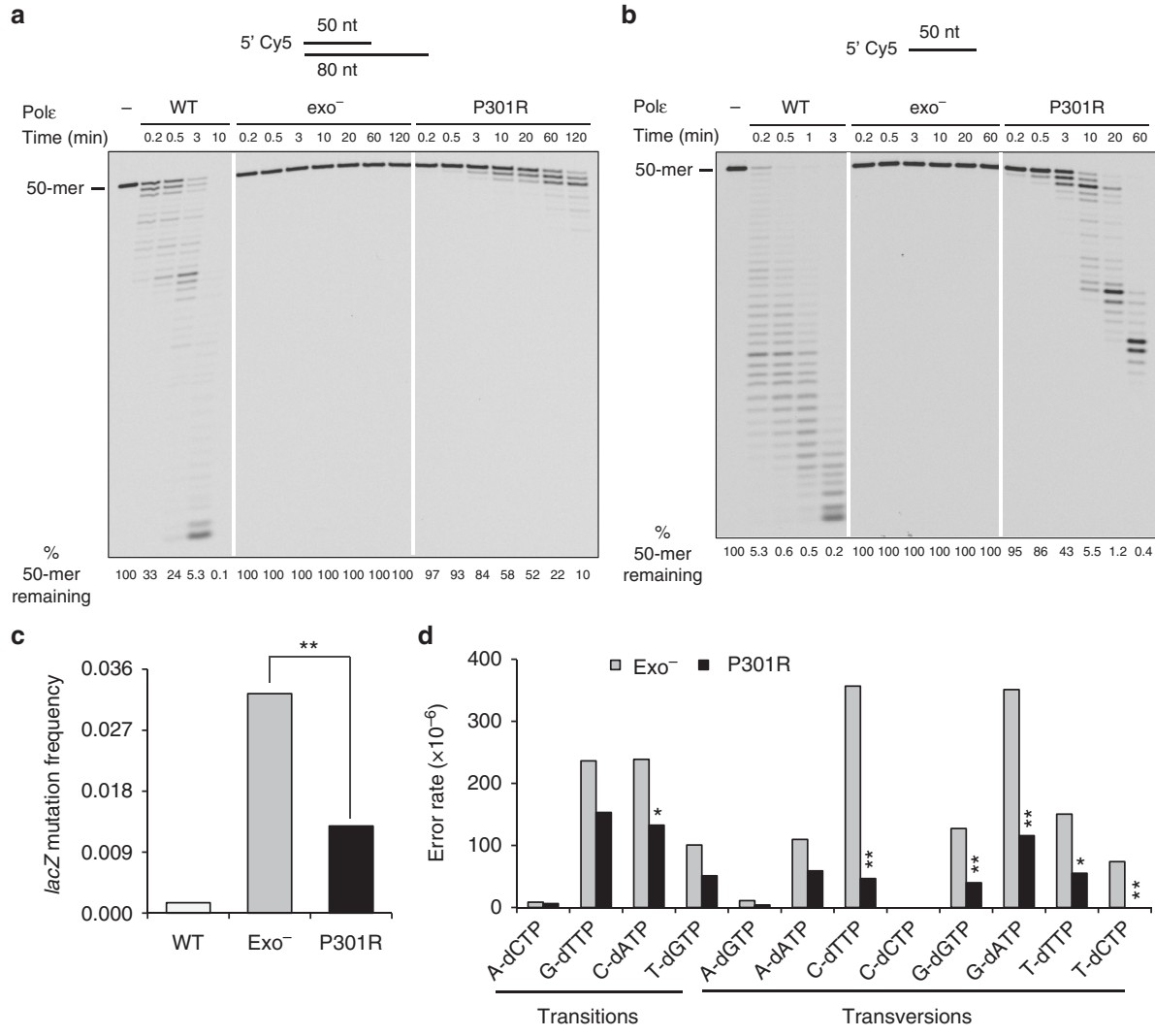

**Fig. 1** Polε-P301R retains weak 3′→5′ exonuclease activity and is more accurate than the proofreading-deficient Polε. **a** Exonuclease activity of wild-type Polε (WT), exo⁻ Polε and Polε-P301R was assayed with 25 nM P50/T80 oligonucleotide substrate and 6.25 nM polymerase. Representative of >10 independent experiments. **b** Exonuclease activity of the Polε variants was assayed with 25 nM P50 single-stranded oligonucleotide and 4 nM polymerase. Representative of six independent experiments. **c** lacZ mutation frequencies resulting from in vitro DNA synthesis by wild-type Polε, exo⁻ Polε and Polε-P301R. **d** In vitro error rates for the 12 possible base-base mispairs generated by exo⁻ Polε and Polε-P301R. Source data for **a** and **b** are provided in a Source Data file. Data for **c** and **d** are from Supplementary Table 1. Asterisks indicate statistically significant differences between exo⁻ Polε and Polε-P301R: *$p < 0.05$; **$p < 0.01$ (Fisher's exact test)

P301R does not have a particularly high error rate, and all of its observed in vitro infidelity may just result from the partial exonuclease defect.

**Ultramutation in vivo results from Polε-P301R errors.** Because the modest mutator properties of the purified Polε-P301R (Fig. 1c, d) were inconsistent with its strong mutator effect in vivo[11], we next asked whether the mutations in the pol2-P301R strains, in fact, resulted from Polε-P301R-mediated DNA synthesis. We previously showed that mutator effects of replicative DNA polymerase variants can be caused by the recruitment of the error-prone translesion synthesis DNA polymerase ζ (Polζ) to stalled replication forks[19]. Deletion of the REV3 gene encoding the catalytic subunit of Polζ did not decrease the mutation rate in the pol2-P301R strains, indicating that Polζ is not responsible for the ultramutation (Fig. 2a). We then examined whether errors occurring in the pol2-P301R strains are subject to correction by MMR and, therefore, are generated during replicative DNA

synthesis. Tetrad dissection of heterozygous POL2/pol2-P301R MSH6/msh6Δ diploids showed that the combination of the P301R substitution with the MMR defect results in synthetic lethality (Fig. 2b, left). The double mutant pol2-P301R msh6Δ cells were able to divide and form microcolonies of varying size before cell division stopped (Fig. 2b, right), a phenotype indicative of replication error catastrophe[3]. In contrast, double pol2-4 msh6Δ mutants carrying exo⁻ Polε and lacking the Msh6-dependent MMR were readily produced by sporulation of POL2/pol2-4 MSH6/msh6Δ diploids (Fig. 2b, middle). The synthetic lethality of pol2-P301R and msh6Δ demonstrates that the pol2-P301R strains accumulate an enormous amount of DNA replication errors, which, in the absence of MMR, exceed the viability threshold. These results also illustrate the much stronger mutator activity of Polε-P301R in vivo, as compared to exo⁻ Polε. To gain further insight into the origin of Polε-P301R-mediated mutations, we compared the spectra of base substitutions accumulating in the pol2-4 and pol2-P301R strains to the mutational specificity of the respective polymerases deduced from the in vitro fidelity assays.

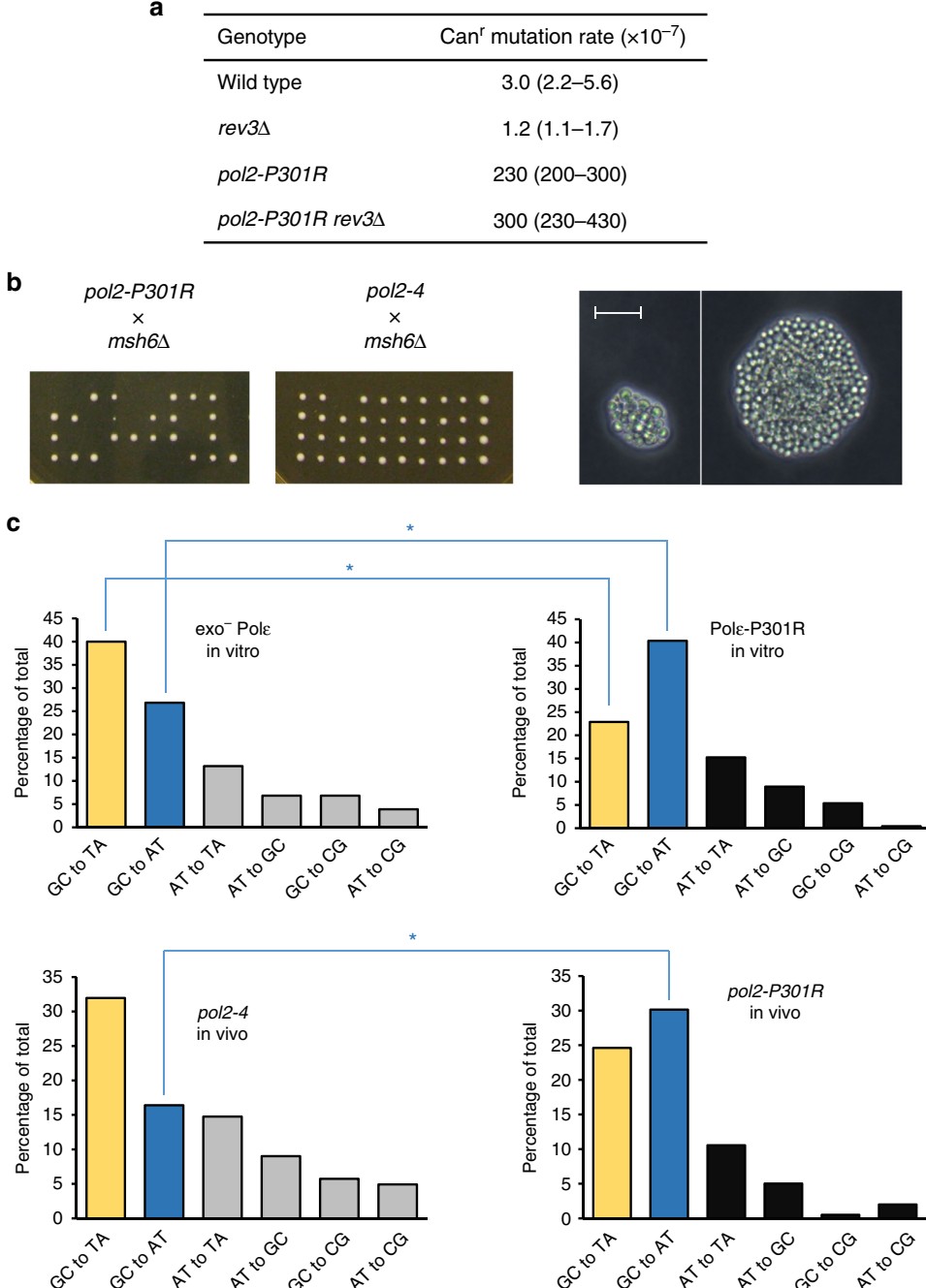

**Fig. 2** In vivo evidence suggests that the ultramutator phenotype of *pol2-P301R* strains results from errors made by Polε-P301R during replicative DNA synthesis. **a** Mutator effect of *pol2-P301R* is not dependent on Polζ. Rate of spontaneous Can$^r$ mutation was measured in haploid *rev3Δ*, *pol2-P301R* and *pol2-P301R rev3Δ* mutants and an isogenic wild-type strain. Mutation rates are given as medians for at least nine cultures with 95% confidence limits in parentheses. Source data are provided in a Source Data file. **b** The *pol2-P301R* shows synthetic lethal interaction with the MMR defect indicative of a replication error catastrophe. Left, tetrad analysis of diploids heterozygous for *pol2-P301R* and *msh6Δ*, and *pol2-4* and *msh6Δ*. While double mutants were readily produced by sporulation of *POL2/pol2-4 msh6Δ/MSH6* diploids, no viable *pol2-P301R msh6Δ* spores were obtained from the *pol2-P301R* x *msh6Δ* cross. Right, two examples of dead cell groups formed upon germination of the *pol2-P301R msh6Δ* haploid spores. Scale bar, 20 μm. Representative of two independent experiments. **c** The error signatures of exo$^-$ Polε and Polε-P301R are apparent in the in vivo mutational spectra of *pol2-4* and *pol2-P301R* strains, respectively. Proportions of individual base substitutions generated during in vitro synthesis by exo$^-$ Polε (top left) and Polε-P301R (top right) were obtained by combining data for the corresponding reciprocal base-base mismatches from Supplementary Table 1. Proportions of individual base substitutions in the *pol2-P301R* yeast strain (bottom right) were determined by DNA sequence analysis of 194 Can$^r$ mutants containing a total of 199 mutations in the *CAN1* gene (Supplementary Table 2). The analogous data for the *pol2-4* yeast strain (bottom left) are from[54]. Asterisks indicate statistically significant differences in the proportions of GC→AT transitions and GC→TA transversions ($p < 0.01$, Fisher's exact test)

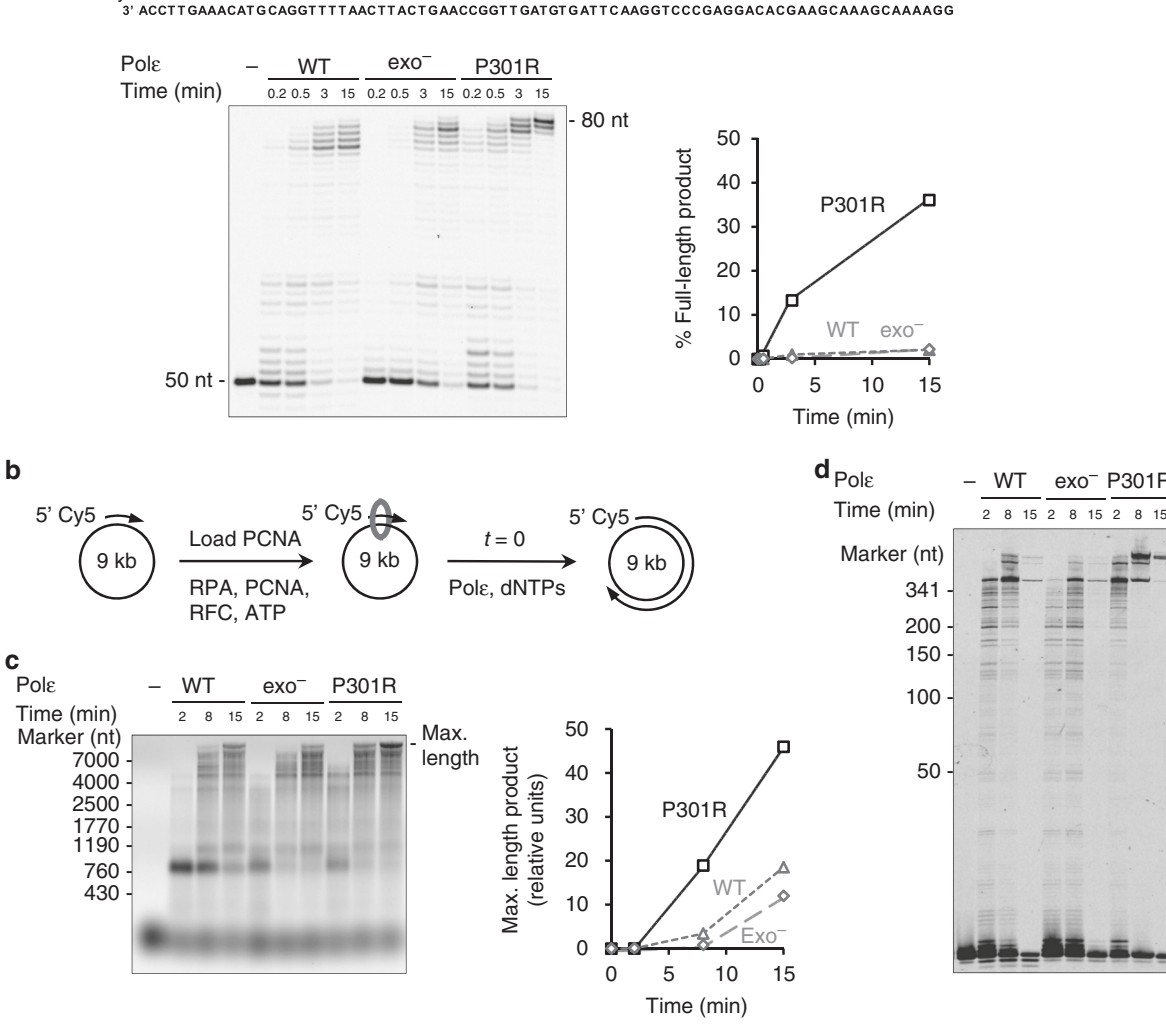

**Fig. 3** Polε-P301R has a robust DNA polymerase activity superior to that of the wild-type or proofreading-deficient Polε. **a** DNA polymerase activity was assayed using 25 nM correctly matched P50/T80a oligonucleotide substrate (shown above the gel image) and 6.25 nM polymerase, and the fraction of full-length product (80 nt) was quantified. Representative of two independent experiments. **b** Schematic of DNA replication assay on the circular M13/CAN1(1-1560-F) substrate. **c** M13/CAN1(1-1560-F) replication reactions were performed at a Polε:substrate ratio of 5:1. The products were separated in a 0.8% alkaline agarose gel, and the fraction of maximal-length products was quantified. Representative of two independent experiments. **d** The M13/CAN1 (1-1560-F) replication reactions were performed at a Polε:substrate ratio of 1:5, and the products were analyzed by 8 M Urea PAGE. One experiment. Source data for **a**, **c**, and **d** are provided in a Source Data file

The mutational spectra produced by purified exo⁻ Polε and Polε-P301R in vitro differ primarily in the proportions of the two most frequent classes of base substitutions, GC→AT transitions and GC→TA transversions (Fig. 2c, top left and top right). The in vivo *pol2-4* spectrum was remarkably similar to the spectrum of mutations resulting from DNA synthesis by exo⁻ Polε in vitro (Fig. 2c, top left and bottom left). At the same time, the in vivo *pol2-P301R* spectrum showed an increase in the GC→AT transition/GC→TA transversion ratio predicted by the in vitro specificity of Polε-P301R (Fig. 2c, top right and bottom right). Thus, the exceptionally strong mutator phenotype of the *pol2-P301R* strains appears to result from replicative DNA synthesis by the Polε-P301R variant. The lower error rate of the purified Polε-P301R in comparison to exo⁻ Polε (Fig. 1) suggests that additional factors must enhance the impact of its infidelity on mutagenesis in vivo.

**Polε-P301R is a hyperactive DNA polymerase**. In a search for additional effects of the P301R substitution, we compared the

DNA polymerase activity of wild-type Polε, exo⁻ Polε and Polε-P301R. In a primer extension assay using an oligonucleotide template (P50/T80a substrate), Polε-P301R had a substantially higher activity in comparison to both wild-type Polε and exo⁻ Polε, as indicated by a greatly increased accumulation of full-length products (Fig. 3a). Active site titration indicated that the fraction of active polymerase was comparable in the wild-type Polε, exo⁻ Polε and Polε-P301R preparations (Supplementary Fig. 5), therefore, the increased synthesis by Polε-P301R was not due to a higher concentration of active enzyme. We next determined if Polε-P301R also showed an enhanced DNA polymerase activity during copying of long natural templates in reactions reconstituted with the auxiliary replication proteins proliferating cell nuclear antigen (PCNA), replication factor C (RFC) and replication protein A (RPA). PCNA was stably loaded on a singly primed 9.0-kb circular single-stranded DNA substrate, M13/CAN1(1-1560-F)[20], and replication reactions were initiated by the addition of Polε (Fig. 3b). Similar to the results with the

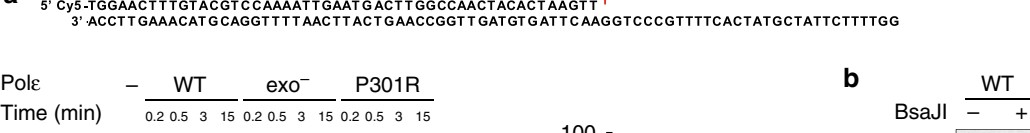

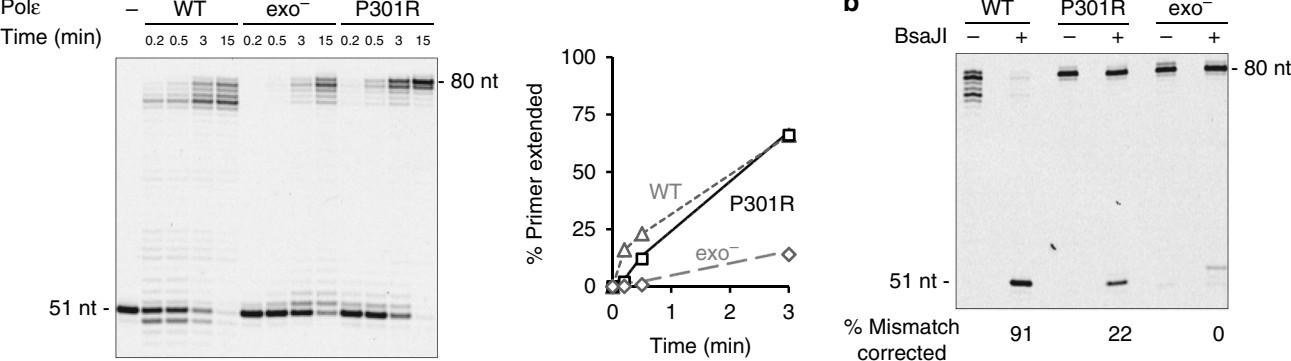

**Fig. 4** Increased mismatch extension capacity of Polε-P301R. **a** DNA polymerase activity was assayed on P51T/T80 oligonucleotide substrate containing a terminal G-T mismatch, and the fraction of primer extended (≥52 nt) was quantified. The polymerase and DNA substrate concentrations are as in Fig. 3a. Representative of seven independent experiments. **b** Polε variants were incubated with the P51T/T80 substrate for 30 min, and relative efficiency of mismatch extension vs. proofreading was determined by BsaJI digestion of the reaction products. The appearance of 51-nt restriction fragment indicates that the mismatch has been corrected by the polymerase. The 80-nt fragments resistant to BsaJI digestion represent products of mismatch extension. Representative of two independent experiments. Source data are provided in a Source Data file

oligonucleotide templates, Polε-P301R decidedly outperformed both wild-type Polε and exo⁻ Polε (Fig. 3c, d). The wild-type Polε was slightly more efficient than exo⁻ Polε at accumulating long products in this assay, as was also previously observed with some DNA substrates[21,22], but Polε-P301R was clearly superior to both of them (Fig. 3c). Separating the reaction products in a sequencing gel showed that DNA synthesis by wild-type Polε and by exo⁻ Polε was impeded at several major pause sites, and Polε-P301R was much more efficient at bypassing these sites (Fig. 3d).

**Increased mismatch extension capacity of Polε-P301R.** We next determined whether Polε-P301R had a higher ability to extend mismatched primer termini. Incorrect nucleotide incorporation must be followed by extension of the aberrant primer terminus in order to result in a mutation. Replicative DNA polymerases are generally poor extenders, which is one of the mechanisms contributing to mutation avoidance. The delay in DNA synthesis caused by the inability to extend a mismatched primer terminus normally provides opportunities for correction of the error by intrinsic or extrinsic proofreading mechanisms[23]. In reactions with an oligonucleotide primer-template substrate containing a terminal G-T mismatch, P51T/T80, Polε-P301R showed a greatly increased DNA synthesis activity in comparison to exo⁻ Polε (Fig. 4a). While it was initially delayed relative to the wild-type Polε that can remove the mismatched nucleotide, Polε-P301R was able to catch up and produce the same amount of extended products as the wild-type Polε during the time course of the reaction. It was also more efficient than exo⁻ Polε at extending primers containing internal mismatches in the vicinity of 3′ terminus (Supplementary Fig. 6). Since Polε-P301R has residual exonuclease activity (Fig. 1a), the observed efficient synthesis on the mismatched substrates could potentially result from the action of the exonuclease followed by extension of the resulting correctly matched primer terminus. To be able to distinguish between a true extension of the mismatch and a correction followed by extension, we designed the P51T/T80 substrate such that the T80 template contained a recognition sequence for the BsaJI restriction endonuclease at the primer-template junction. Incorporation of the mismatched 3′-terminal T of the primer into the reaction product would destroy the restriction site. If the polymerase excised the mismatched T before extending the

primer, the products would be digested by BsaJI. Restriction analysis of full-length extension products showed that Polε-P301R excised the mismatched T in 22% of cases, while 78% of products resulted from extension of the abnormal primer terminus (Fig. 4b). In contrast, the wild-type Polε corrected the mismatch in 91% of cases, and only 9% of products resulted from direct extension. As expected, no excision occurred in reactions with exo⁻ Polε. A faint band at the 52-nt position likely resulted from slippage of the primer terminus and incorporation of an additional T across from the upstream A's in the template, followed by extension of the slipped intermediate. Overall, these results indicate that Polε-P301R strongly prefers to extend rather than correct mismatched primer termini, and it greatly surpasses both wild-type Polε and exo⁻ Polε in the extension capacity.

**Increased bypass of hairpin DNA structures by Polε-P301R.** Unusual DNA secondary structures, such as hairpins, cruciforms, G-quadruplex, triplex, and Z-DNA present obstacles for DNA replication machinery[24]. Inverted repeats capable of forming hairpin structures are particularly common, with many repeats present in every gene. Hairpin extrusion is facilitated by the unwinding of duplex DNA during replication, thus, DNA polymerases often encounter such structures. We previously observed that hairpins with a stem as short as 4–6 nucleotides can significantly impede synthesis by replicative DNA polymerases in vitro[20]. The increased DNA polymerase activity of Polε-P301R on long natural templates could be, in part, due to a more efficient bypass of non-B DNA structures. We compared the efficiency of DNA synthesis by wild-type Polε, exo⁻ Polε, and Polε-P301R on the P50/T80H substrate containing 6-nt inverted repeats in the template region. The putative hairpin in this substrate would be located 3-nt downstream of the primer terminus. The wild-type Polε was nearly completely blocked by the hairpin and used its exonuclease activity to degrade the primer (Fig. 5a). The exo⁻ Polε was significantly inhibited but was able to produce a substantial amount of full-length product at later time points (Fig. 5a). This is consistent with a previous report showing that Polε becomes capable of strand displacement once its exonuclease activity is disrupted[21]. Polε-P301R, however, showed dramatically increased hairpin bypass activity in comparison to exo⁻ Polε (Fig. 5a).

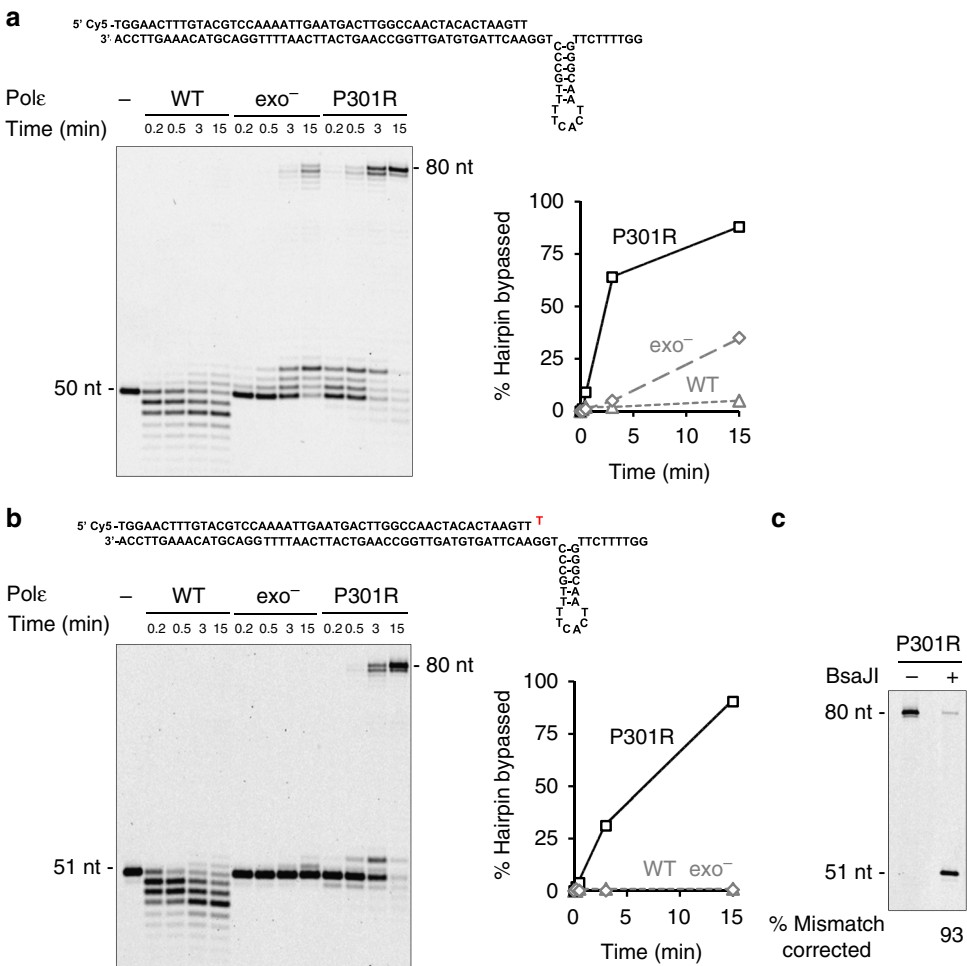

**Fig. 5** Increased bypass of hairpin DNA structures by Polε-P301R. The polymerase and DNA substrate concentrations are as in Fig. 3a. **a** DNA polymerase activity was assayed on P50/T80H oligonucleotide substrate containing 6-bp inverted repeats in the template region, and the fraction of products longer than 71 nt indicating hairpin bypass was quantified. Representative of three independent experiments. **b** DNA polymerase activity was assayed on P51T/T80H oligonucleotide substrate containing inverted repeats in the template region and a terminal G-T mismatch, and the fraction of products longer than 71 nt indicating hairpin bypass was quantified. Representative of four independent experiments. **c** Polε-P301R was incubated with the P51T/T80H substrate for 30 min, and relative efficiency of mismatch extension *vs.* proofreading was determined by BsaJI digestion of the reaction products as in Fig. 4b. One experiment. Source data are provided in a Source Data file

Next, we engineered an even more challenging DNA substrate, P51T/T80H, containing the hairpin-forming inverted repeats in the template region and a mismatched primer terminus. Synthesis by both wild-type Polε and exo⁻ Polε was completely blocked by this double obstacle, but Polε-P301R still efficiently produced full-length products, providing, perhaps, the best illustration of its remarkable power as a DNA polymerase (Fig. 5b). Quantitative analysis of BsaJI digestion showed that Polε-P301R corrected most of the mismatched primer termini before extending them (Fig. 5c), which could be expected given the impeding effect of the hairpin on polymerization.

## Discussion

The exceptionally strong mutator effect of the human Polε-P286R variant modeled in yeast suggested functional alterations beyond a simple loss of proofreading, but the nature of these additional alterations remained elusive. The high recurrence of Polε-P286R in tumors and the scarcity of mutations that produce catalytically inactive Polε implies that these additional consequences of the arginine substitution may be responsible for its pathogenicity.

The present study shows that, despite the catastrophic rate of replication errors in vivo, purified yeast mimic of Polε-P286R is not remarkably inaccurate. It is more accurate than the exonuclease-deficient Polε and has some proofreading capability. However, a major property that distinguishes the cancer-associated variant from both wild-type and exonuclease-deficient Polε is an abnormally high DNA polymerase activity. Increased activity was observed in all assays used, in the absence and in the presence of accessory proteins, and was particularly impressive with mismatched and secondary structure-containing substrates that generally impede synthesis by replicative DNA polymerases.

A companion study by Parkash et al.[14] describes the crystal structure of yeast Polε-P301R that may provide a rationale for these unusual properties. In this structure, the side chain of Arg301 dwells in the space that must be occupied by the 3′-terminal nucleotide of the primer when Polε is in the editing mode. The arginine substitution also affects metal binding at the exonuclease active site and coordination of the catalytic residue E292. In addition to adverse effects on catalysis, these changes would likely prevent proper positioning of the primer terminus in

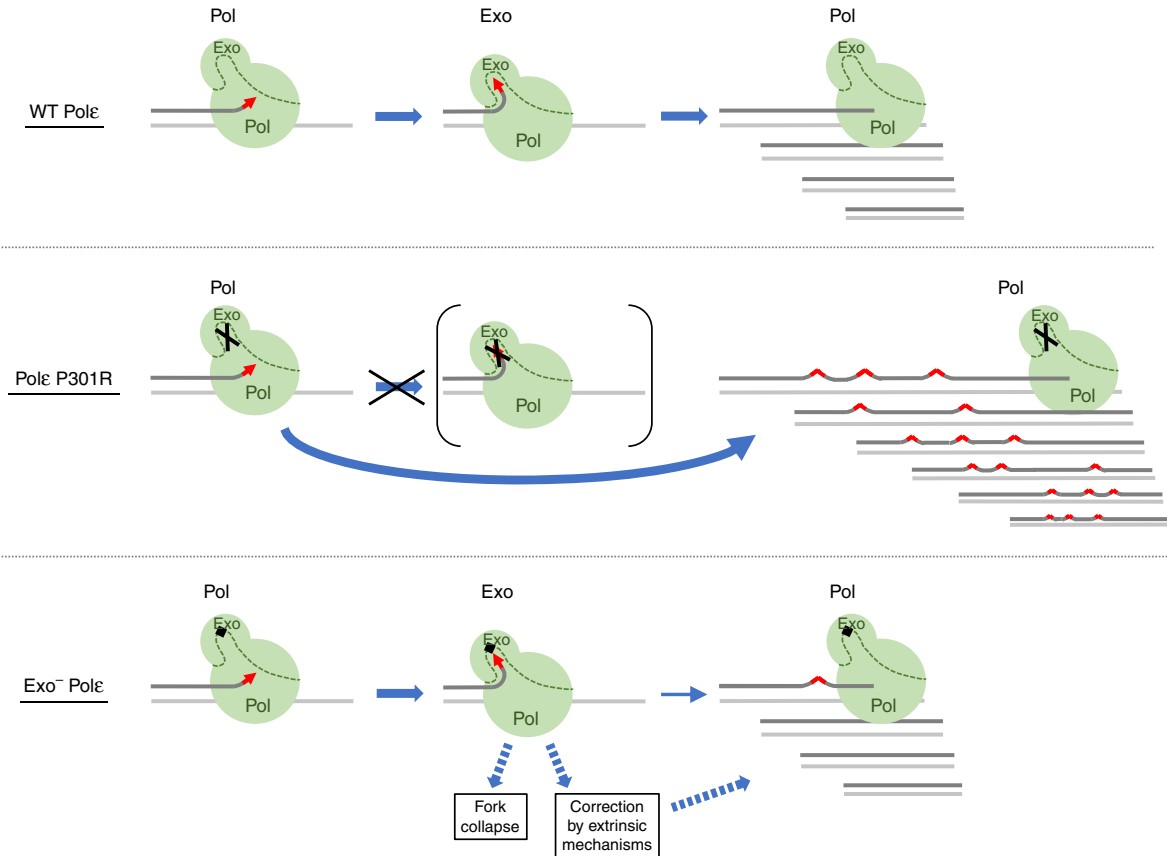

**Fig. 6** Polε exonuclease domain alterations as a source of increased DNA polymerase activity and ultramutation in cancers. Synthesis by both wild-type Polε and exo⁻ Polε involves partitioning of the primer terminus between the polymerase (Pol) and exonuclease (Exo) active sites. This partitioning limits the rate of DNA synthesis and prevents efficient mismatch extension, leading to the correction of errors by the intrinsic (wild-type Polε) or extrinsic (exo⁻ Polε) mechanisms. In contrast, the cancer-associated P301R substitution restricts access of the primer terminus to the exonuclease site, prompting the polymerase to stay in the elongation mode and, thus, resulting in hyperactivity and a high mutation rate

the exonuclease site. In contrast, the catalytic residue mutation in exo⁻ Polε prevents hydrolysis but does not create steric hindrance for the movement of the 3′-end to the exonuclease site[25]. This appears to be the key difference between the two enzymes, as no changes were seen in the structure of the DNA polymerase domain. The model in Fig. 6 integrates our findings with the structural data to explain how the local structural alteration in the exonuclease domain could lead to an increased DNA polymerase activity and ultramutator phenotype. Insertion of a non-complementary nucleotide by the wild-type Polε (Fig. 6, top) inhibits further elongation and promotes transfer of the primer terminus to the exonuclease active site. Removal of the mismatched nucleotide allows Polε to resume DNA synthesis and generate predominantly error-free products. This is consistent with the established role of exonucleolytic proofreading in enhancing the fidelity of replicative DNA polymerases[23], and, indeed, we observed that the wild-type Polε prefers correction >10-fold over extension when presented with a mismatched primer terminus (Fig. 4b). In the case of Polε-P301R (Fig. 6, middle), we propose that the inability to accommodate single-stranded DNA in the exonuclease site forces the enzyme to stay in the polymerization mode, resulting in higher activity, better mismatch extension, and ultimately faster DNA synthesis with the majority of errors converted into mutations. In contrast, exo⁻ Polε (Fig. 6, bottom), while being unable to proofread, still allows partitioning of the DNA between the polymerase and exonuclease sites. This partitioning likely makes exo⁻ Polε slower than Polε-P301R and similar to the wild-type Polε in terms of the overall

rate of elongation. However, a nucleotide misinsertion would severely impede further synthesis by exo⁻ Polε, prompting it to either remain bound in the editing mode or dissociate. In vitro fidelity assays allow substantial time for extension and multiple binding events, revealing that exo⁻ Polε has the potential to generate more mutations than the partially exonuclease-proficient Polε-P301R (Fig. 1c, d). We propose that this potential is not realized in the context of a rapidly moving replication fork in vivo, and the poor mismatch extension ability of exo⁻ Polε results in aborted replication products or correction of the mismatch by extrinsic mechanisms (Fig. 6, bottom). This model implies that the ultramutator effect of Polε-P301R results from efficient extension of mismatches formed by canonical nucleotides. In addition, the increased polymerase activity of this Polε variant may facilitate mutagenic bypass of endogenous DNA lesions, a possibility that could be tested in future studies.

While P286R is the most frequently seen variant in tumors, many other recurrent variants affect amino acid residues at the DNA binding interface in the exonuclease domain[26]. When modeled in yeast, the vast majority confer mutator effects exceeding the effects of exonuclease deficiency[13]. It is tempting to suggest that these variants, too, limit the ability of Polε to accommodate DNA in the exonuclease site and result in an increased DNA polymerase activity. At the same time, the rarity of mutations at catalytic residues in the exonuclease domain in cancers could reflect the fact that they do not prevent sliding of the primer terminus to the exonuclease site and, thus, do not provide Polε with the robustness needed to acquire the

ultramutator phenotype. It is interesting to note that studies of other DNA polymerases, such as Polδ and T4 DNA polymerase, have identified amino acid changes in the proofreading domains that impair the ability to switch between polymerase and exonuclease sites, but these mutants do not show increased polymerase activity[27,28]. The unique properties of Polε-P301R suggest that the coordination of polymerase and exonuclease activities is different in Polε, and eliminating the option to bind in the exonuclease mode makes Polε a much more efficient polymerase. While this may be the change selected for during tumorigenesis, it remains to be established how the unusual way of balancing the two catalytic activities facilitates the functions of wild-type Polε in DNA replication and other cellular transactions.

According to the currently accepted eukaryotic replication fork model, Polε is primarily responsible for synthesis of the leading DNA strand, including both polymerization and proofreading of errors, while the second replicative polymerase with a $3' \rightarrow 5'$ exonuclease activity, Polδ, synthesizes most of the lagging strand[29]. This view is supported by a multitude of studies, including strand-specific increases in mutagenesis in cells with inaccurate Polε and Polδ variants[30–33], strand-specific ribonucleotide incorporation in cells with Polε and Polδ variants deficient in ribonucleotide discrimination[34,35], a clear role of Polδ and not Polε in the proofreading of errors made by Polα[36] and in Okazaki fragment maturation[37,38], and the cooperation of Polε and not Polδ with the leading strand helicase in reconstituted in vitro replication reactions[39]. It is important to note that all available in vivo evidence for the primary role of Polε as a leading-strand polymerase is based on studies of mutants with altered nucleotide selection or proofreading. In *Saccharomyces cerevisiae*, the *pol2-4* allele encoding exo⁻ Polε and the *pol2-M644G* allele affecting nucleotide selectivity have been used to deduce the roles of exonuclease and DNA polymerase activities of Polε[32–34]. The critical assumption in these studies was that the mutant variants correctly reflect the function of wild-type Polε. This assumption relies on the biochemical evidence that the DNA polymerase activity and processivity of the mutant variants is similar to those of the wild-type Polε[32,40,41]. However, the finding that Polε-P301R possessing higher activity causes a two-orders-of-magnitude stronger mutator effect in vivo than exo⁻ Polε or Polε-M644G[11] suggests that the contribution of Polε to DNA replication can be increased well beyond what these previously studied mutants detected. Our data suggest two possibilities. First, the previously studied variants (exo⁻ Polε, Polε-M644G), and by inference the wild-type Polε, do not replicate the entire leading strand but rather contribute at a small percentage of replication forks or in a small percentage of nucleotide incorporation events. The two-orders-of-magnitude difference in the mutator effects of Polε-P301R and the other variants suggests that this fraction could be as small as 1%. Indeed, Polδ can replicate both leading and lagging strands during SV40 origin-dependent replication in vitro[42]. It has also been suggested that Polε contributes little to chromosomal DNA replication in vivo, with Polδ primarily synthesizing both strands[43], but the only attempt to prove this experimentally[44] has been inconclusive[45]. The second possibility is that Polε might replicate the majority of the leading strand, but, because of its poor mismatch extension capacity, only a tiny proportion of its errors result in mutations. Errors that Polε cannot proofread itself (nearly all errors in the case of exo⁻ Polε) would be corrected by extrinsic mechanisms or result in incomplete replication products, as illustrated in Fig. 6. Genetic evidence suggests that the exonuclease activity of Polδ can correct errors made by Polε[43,46], but other cellular nucleases could possibly also help remove a poorly extendable primer terminus. The P301R substitution may be changing this arrangement and greatly increasing the contribution of Polε to DNA replication

and/or mutagenesis. Further studies of this and other cancer-associated Polε variants will not only provide insight into the molecular pathogenesis of ultramutated tumors but will also help define the mechanisms that normally regulate the cellular function of Polε.

## Methods

**Saccharomyces cerevisiae strains and plasmids.** The haploid strain FM113 (*MAT*a *ura3-52 trp1-289 leu2-3,112 prb1-1122 prc1-407 pep4-3*)[47] and its *pol2-4* and *pol2-P301R* derivatives were used to overproduce and purify wild-type Polε, exo⁻ Polε and Polε-P301R, respectively. Strains used for genetic experiments are isogenic to E134[48]. The *pol2-4* and *pol2-P301R* mutants of all strains were constructed by replacing the chromosomal *POL2* gene with the mutant alleles using plasmids YIpJB1[15] and YIpDK1[11]. DK028/029 and DK007 are *pol2-P301R* and *pol2-4* mutants, respectively, of the haploid strain TM44 (*MAT*α *ade5-1 lys2-InsE*$_{A14}$*trp1-289 his7-2 leu2-3,112 ura3-52 can1Δ::loxP*)[17]. TM41 (*MAT*a *ade5-1 lys2-Tn5-13 trp1-289 his7-2 leu2-3,112 ura3-4 CAN1::Kl.LEU2 msh6Δ::KanMX*) was constructed by T. M. Mertz in the Shcherbakova laboratory by disrupting the *MSH6* gene in TM30 strain[17] with a PCR-amplified *KanMX* cassette[49]. TM41 was crossed to DK029 and DK007 to generate diploids (DK517 and DK518, respectively) heterozygous for the *pol2* and *msh6* mutations. The haploid strain DK004 (*MAT*α *ade5-1 lys2-InsE*$_{A14}$*trp1-289 his7-2 leu2-3,112 ura3-4 CAN1::Kl.LEU2 pol2-P301R*)[11] was used for the mutational spectra analysis. The haploid strains CB29 and CB30 are *pol2-P301R* mutants of OK29 (*MAT*α *ade5-1 lys2::InsE*$_{A14}$*trp1-289 his7-2 leu2-3,112 ura3-G764A-LEU2*) and its *rev3Δ::KanMX* variant[50], respectively. OK29, OK29 *rev3Δ::KanMX*, CB29 and CB30 were used to study the genetic interaction of *pol2-P301R* and *rev3* mutations.

Plasmids pJL1 and pJL6 for overproduction of the four subunits of yeast Polε[51] were kindly provided by Erik Johansson (Umeå University, Sweden). The *pol2-4* and *pol2-P301R* mutations were introduced into the *POL2* gene in pJL1 by site-directed mutagenesis.

**Proteins.** Untagged wild-type Polε, exo⁻ Polε, and Polε-P301R were purified by conventional chromatography from yeast strains overproducing all four Polε subunits using an adaptation of the previously described procedure[51]. The purification buffers were as follows: buffer A contained 150 mM Tris-acetate, pH 7.8, 50 mM sodium acetate, 2 mM EDTA, 1 mM EGTA, 10 mM NaHSO₃, 1 mM dithiothreitol, 5 μM pepstatin A, 5 μM leupeptin, 0.3 mM phenylmethylsulfonyl fluoride, and 5 mM benzamidine; buffer B contained 25 mM Hepes-NaOH, pH 7.6, 10% glycerol, 1 mM EDTA, 0.5 mM EGTA, 0.005% Nonidet P-40, 1 mM dithiothreitol, 5 μM pepstatin A, 5 μM leupeptin, 5 mM NaHSO₃, and sodium acetate at a concentration (mM) indicated by the subscript number (for example, buffer B$_{50}$ is buffer B with 50 mM sodium acetate); and buffer C contained 25 mM Hepes-NaOH, pH 7.6, 10% glycerol, 1 mM EDTA, 0.005% Nonidet P-40, 400 mM sodium acetate, 5 mM dithiothreitol, 5 mM NaHSO₃, 2 μM leupeptin, and 2 μM pepstatin A. Approximately 100 g of wet cells were harvested from 20 L of culture medium and resuspended in 36 ml ddH₂O. Cells were opened by Spex SamplePrep 6870 Freezer/Mill (SPEX SamplePrep, USA). The volume of cell extract was measured, and 5× buffer A stock and ammonium sulfate were added to final concentrations of 1× buffer A and 175 mM ammonium sulfate. 0.4 ml of 10% PEI, pH 7.9 was then added dropwise per 10 ml of cell extract, and the extract was stirred on the ice for 15 min, followed by centrifugation at 39,000×*g* for 30 min at 4 °C. Next, 2.8 g of solid ammonium sulfate was added per 10 mL of supernatant, dissolved by stirring on ice for 45 min, and proteins were precipitated by centrifugation at 39,000×*g* for 30 min at 4 °C. The precipitate was resuspended in 50 mL of buffer B$_{50}$, and 1.06 g of solid ammonium sulfate was added per 10 mL of sample, followed by stirring on ice for 45 min and centrifugation at 39,000×*g* for 30 min at 4 °C. Then, 0.55 g of solid ammonium sulfate was added per 10 mL of supernatant, followed by stirring on ice for 45 min and centrifugation at 39,000×*g* for 30 min at 4 °C. The Polε-enriched precipitate was resuspended in 50 mL of buffer B$_{50}$ and frozen. The following day, the sample was dialyzed against 2 L of buffer B$_{50}$ for 2 h and centrifuged at 39,000×*g* for 30 min at 4 °C. The supernatant was loaded onto a 20-mL SP column (GE, USA) equilibrated with B$_{200}$, the column was washed with B$_{200}$, and proteins were eluted with B$_{750}$. The SP fractions were loaded onto a 5-mL HiTrap Q column (GE, USA) equilibrated with B$_{500}$, the column was washed with B$_{200}$, and proteins were eluted with a 40-mL linear gradient from B$_{200}$ to B$_{1200}$. The HiTrap Q fractions were diluted with buffer B$_0$ to a final sodium acetate concentration of 100 mM. The samples were loaded onto a Mono S column (GE, USA) equilibrated with B$_{100}$, and proteins were eluted with a 20-mL linear gradient from B$_{100}$ to B$_{1200}$. The Mono S fractions were concentrated to a final volume of 200 μL by spinning in Amicon Ultra-0.5 mL 3 K Centrifugal Filters (Millipore, USA) in 40° fixed angle rotor at 19,000×*g* at 4 °C and loaded onto Superdex 200 10/300 GL filtration column (GE, USA) equilibrated with buffer C. The gel filtration fractions were aliquoted and stored at −80 °C.

The preparation of yeast PCNA used in this work has been described[17]. To purify yeast RPA, *E. coli* strain BL21(DE3) was transformed by the expression vector p11d-tRPA[52], grown to OD$_{600}$ of 0.6 at 37 °C, and induced by 0.4 mM IPTG for 2 h. RPA was then purified using a 10-mL Affi-Gel Blue column (Bio-Rad), a

HAP column (Bio-Rad), and a Mono-Q(HR5/5) column (GE). Yeast RFC was kindly provided by Peter Burgers (Washington University School of Medicine).

**Exonuclease and polymerase assays on oligonucleotides**. Substrates for DNA polymerase and exonuclease assays were prepared by annealing Cy5-labeled oligonucleotides P50 (Cy5-5′-TGGAACTTTGTACGTCCAAAATTGAATGACTTG GCCAACTACACTAAGTT-3′) or P51T (Cy5-5′-TGGAACTTTGTACGTCCAAA ATTGAATGACTTGGCCAACTACACTAAGTTT-3′) to 80-mer templates T80a (5′-GGAAAACGAAACGAAGCACAGGAGCCCTGGAACTTAGTGTAGTTGG CCAAGTCATTCAATTTTGGACGTACAAAGTTCCA-3′) or T80 (5′-GGTTTT CTTATCGTATCACTTTTGCCCTGGAACTTAGTGTAGTTGGCCAAGTCATT CAATTTTGGACGTACAAAGTTCCA-3′) containing an AGTTCCA restriction site sequence (underlined), or T80H (5′-GGTTTTCTT**GGGCAA**TCACTT**TTG CCC**TGGAACTTAGTGTAGTTGGCCAAGTCATTCAATTTTGGACGTA CAAAGTTCCA-3′) containing the BsaJI recognition sequence and 6-nt inverted repeats (highlighted in bold). The annealing was performed by incubating the primer and template at a ratio of 1:1 in the presence of 150 mM NaAc at 92 °C for 2 min and then cooling slowly to room temperature (~2 h). Exonuclease activity was also analyzed using the single-stranded P50 oligonucleotide alone. For exonuclease assays, the 10-μL reaction contained 40 mM Tris-HCl pH 7.8, 1 mM dithiothreitol, 0.2 mg mL$^{-1}$ bovine serum albumin, 8 mM MgAc$_2$, 125 mM NaAc, 25 nM oligonucleotide substrate, and Polε at the indicated concentration. For DNA polymerase assays, the reactions additionally contained dNTPs at their intracellular S-phase concentrations (30 μM dCTP, 80 μM dTTP, 38 μM dATP, and 26 μM dGTP)[17]. The samples were incubated at 30 °C for the times indicated. For BsaJI restriction digestion, the samples were desalted by centrifugation through G50 microspin columns (GE Healthcare) and incubated with BsaJI at 60 °C for 1 h. The reactions were quenched by the addition of an equal volume of 2× loading buffer containing 95% deionized formamide, 100 mM EDTA, and 0.025% Orange G. After boiling for 3 min and cooling on ice, 6-μL samples were subjected to electrophoresis in 10% denaturing polyacrylamide gel containing 8 M urea in 1× TBE. Quantification was done by fluorescence imaging on a Typhoon system (GE Healthcare).

**Replication assays on M13/CAN1(1-1560-F) substrate**. Singly primed circular DNA substrates for DNA polymerase assays were prepared by annealing the Cy5-labeled oligonucleotide P50-M13 (Cy5-5′-AAGGAATCTTTGTGAGAAAACTGT GAAAGAGGATGTAACAGGGATGAATG-3′) to the M13/CAN1(1-1560-F) single-stranded DNA[20] as described above. For analysis by alkaline agarose gel electrophoresis, 10-μL replication reactions contained 40 mM Tris-HCl pH 7.8, 8 mM MgAc$_2$, 125 mM NaAc, 1 mM dithiothreitol, 0.2 mg mL$^{-1}$ bovine serum albumin, 1 mM ATP, dNTPs at the intracellular S-phase concentrations (see previous subsection), 20 nM singly primed M13/CAN1(1-1560-F), 7.5 μM RPA, 2 nM RFC, 20 nM PCNA, and 100 nM wild-type Polε, exo⁻ Polε or Polε-P301R. For analysis in sequencing gel, 30-μL reactions contained 40 mM Tris-HCl pH 7.8, 8 mM MgAc$_2$, 125 mM NaAc, 1 mM dithiothreitol, 0.2 mg mL$^{-1}$ bovine serum albumin, 1 mM ATP, dNTPs at the intracellular S-phase concentrations, 20 nM singly primed M13/CAN1(1-1560-F), 7.5 μM RPA, 2 nM RFC, 20 nM PCNA, and 4 nM wild-type Polε, exo⁻ Polε or Polε-P301R. RPA was the first protein added, followed by a 1-min incubation at 30 °C, then RFC and PCNA were added followed by another 1-min incubation at 30 °C, and then the replication was initiated by the addition of Polε. Reactions were stopped by the addition of 1 μL of 500 mM EDTA and 1 μL of 2% sodium dodecyl sulfate (SDS), incubated with 2 μL of 20 mg mL$^{-1}$ Proteinase K (ThermoFisher Scientific) at 55 °C for 1 h and purified by phenol/chloroform extraction. For alkaline agarose gel electrophoresis, 10-μL samples were mixed with 2 μL of 6× alkaline loading buffer containing 300 mM NaOH, 6 mM EDTA, 18% (w/v) Ficoll, 0.15% (w/v) bromocresol green, and 0.25% (w/v) xylene cyanol, and the reaction products were separated in 0.8% alkaline agarose gel. For sequencing gels, DNA from 30-μL samples was precipitated by ethanol and dissolved in 6 μL of 2× loading buffer containing 95% deionized formamide, 25 mM EDTA, and 0.025% Orange G. After boiling for 3 min and cooling on ice, the samples were subjected to electrophoresis in 10% denaturing polyacrylamide gel containing 8 M urea in 1× TBE. Quantification was done by fluorescence imaging on a Typhoon system (GE Healthcare).

**In vitro DNA synthesis fidelity**. Double-stranded M13mp2 substrate with a 407-nucleotide single-stranded region was prepared by annealing single-stranded M13mp2 DNA to 6.8-kb PvuII fragment of double-stranded M13mp2 DNA[16,17] and gel-purified. DNA synthesis reactions (25 μL) contained 40 mM Tris-HCl (pH 7.8), 8 mM MgAc$_2$, 125 mM NaAc, 1 mM dithiothreitol, 0.2 mg mL$^{-1}$ bovine serum albumin, 0.5 mM ATP, dNTPs at the intracellular S-phase concentrations, 1 nM gapped substrate, 200 nM RPA, 8 nM RFC, 20 nM PCNA, and 6.25 nM wild-type Polε, exo⁻ Polε or Polε-P301R. The order of protein addition was the same as in M13/CAN1(1-1560-F) replication assays. The reactions were incubated at 30 °C for 10 min or 15 min and stopped by placing the tubes on ice and adding 1.5 μL of 0.5 M EDTA. The efficiency of gap filling was monitored by agarose gel electrophoresis. Transformation of E. coli with the reaction products, scoring of mutant plaques, single-stranded DNA isolation from purified plaques, DNA sequencing and error rate calculation were as previously described[16,41]. All data are based on analysis of lacZ mutants from at least two independent gap-filling reactions.

**In vivo mutation rate and spectrum**. The rate of spontaneous Can$^r$ mutation was measured by fluctuation analysis using at least two independently constructed strains of each genotype. Nine to eighteen 7-mL cultures were started for each strain from single colonies and grown to the stationary phase in liquid yeast extract peptone dextrose medium supplemented with 60 mg/L adenine and 60 mg/L uracil (YPDAU). Cells were plated after appropriate dilutions onto synthetic complete medium containing L-canavanine (60 mg/L) and lacking arginine (SC + CAN) for Can$^r$ mutant count and onto synthetic complete (SC) medium for viable count. Can$^r$ mutant frequency was calculated by dividing the Can$^r$ mutant count by the viable cell count. Mutation rate was calculated from mutant frequency by using the Drake equation[53]. The significance of differences in the mutation rate was assessed by using Wilcoxon–Mann–Whitney non-parametric test. For the mutational spectra determination, independent colonies of the pol2-P301R strain were streaked on YPDAU plates, grown for two days at 30°, and replica-plated onto SC + CAN medium to select for can1 mutants. One Can$^r$ colony was picked from each patch, and the CAN1 gene was amplified by PCR and Sanger-sequenced.

## Data availability
All data used to reach the conclusions are presented fully within the Article and the Supplementary material, and available from the corresponding author upon reasonable request. A Reporting Summary is available as a Supplementary Information file. The source data underlying Figs. 1a, b, 2a, 3a, c, d, 4a, b, 5a–c and Supplementary Figs. 1, 2b, c, 3, 5a, b and 6a–f are provided as a Source Data file.

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

## Acknowledgements

We thank Erik Johansson for pJL1 and pJL6 plasmids, Peter Burgers for RFC, Krista Brown for technical assistance, and Stephanie Barbari and Youri Pavlov for critically reading the manuscript. This work was supported by the National Institutes of Health grant ES015869 and by Nebraska Department of Health and Human Services grant LB506 to PVS, and by the Swedish Cancer Society and the Swedish Research Council grants to AC. C.R.B. was supported by a University of Nebraska Medical Center Graduate Studies Research Fellowship.

## Author contributions

P.V.S. conceived and supervised the study. X.X. purified Polε variants and performed all biochemical assays. D.P.K. contributed to the initial biochemical characterization of Polε variants and studied the genetic interaction with MMR deficiency. C.R.B. studied the genetic interaction with Polζ deficiency. D.P.K. and E.A.M characterized the mutational specificity of pol2-P301R strains. S.S. measured dNTP pools in the pol2-P301R mutants. A.C. supervised the analysis of dNTP pools. X.X. and P.V.S. wrote the manuscript, with input from all other authors.

## Additional information

**Competing interests:** The authors declare no competing interests.

