## [Peer Review File · Nature Communications]

Reviewers' comments:

Reviewer #1 (Remarks to the Author):

In "A recurrent cancer-associated substitution in the exonuclease domain of DNA polymerase produces a hyperactive enzyme", the authors describe biochemical studies aimed at understanding the pathogenic mechanism of Pol-P286R, a Pol variant frequently associated with ultramutated tumors. Pol, a eukaryotic replicative polymerase harbors both DNA synthesis and 3'-5' exonuclease proofreading activity. Exonuclease variants of Pol harboring mutations such as P286R/S/H in its exonuclease domain are frequently associated with ultramutated tumors.

Through a series of well-designed and elegant experiments with a variety of synthetic substrates, the authors shown convincingly that (i) the yeast equivalent of human Pol-P286R has residual exonuclease activity, and (ii) exhibits robust DNA synthesis activity relative to both wild type and the exonuclease dead mutant. Based on these observations and molecular dynamics analysis reported in the companion manuscript, the authors argue that P286R mutation sterically restricts the access of the primer 3'-terminus to the exonuclease active site, effectively shifting the balance towards binding to the polymerase site. Enhanced binding to the polymerase active site leads to accelerated DNA polymerase activity and concomitantly, a dramatic increase in mutations. This proposed mechanism by which P286R variant causes hypermutation is in contrast to the prevailing idea that the pathogenicity of Pol exonuclease variants results solely from a direct disruption of the exonuclease/proofreading activity

No data is reported for the P286S/H mutations, the two other Pol variants commonly associated with ultramutated tumors, and the steric mechanism by which P286R is proposed to restrict access of the DNA primer to the exonuclease site is unlikely to apply to the smaller P286S and P286H mutations. Nonetheless, the work is technically sound and elegant, and the proposed model greatly enhances our understanding of the in vivo ultramutator effect of Pol-P286R. As such, it is suitable for publication in Nature Communications with some modifications throughout the text, and restructuring the Discussion section to eliminate inferences not directly supported by this study.

Comments:

(i) On Pg. 3, the authors note that "...the idea that the pathogenicity of Pol variants results from adverse effects...". This should be changed to "...the idea that the pathogenicity of all Pol exonuclease variants result solely from adverse effects..."

(ii) Pg. 6, Fig. 1a and 1b: What happens to the exonuclease reactions at time periods longer than 10 min and whether the variant enzyme is able to catch up with the WT protein? Authors should consider including exonuclease activity from mismatched template-primer termini and internal mismatches.

(iii) On Pg. 10, "Since Pol-P301R has limited exonuclease..." should be changed to "Since Pol-P301R has residual exonuclease..."

(iv) Pg. 9. The authors report the ability of Pol-P301R to extend from mismatched primer-template termini. For the sake of completion, it is strongly recommended that the authors include DNA synthesis activity from template-primer duplexes harboring mismatches downstream of the primer terminus.

(v) Pg. 13-14: The authors suggest that "...inability to accommodate single stranded DNA in the exonuclease site forces the enzyme to stay in the polymerization mode, ...". This should be rephrased

since there are two possibilities – the enzyme to stay in the polymerization mode or to release the template-primer duplex.

(vi) Pg. 13-14: While the biochemical studies reported here are technically sound, interpretation of these results are dependent on the molecular dynamics analysis of the P301R variant with single stranded DNA modeled at the exonuclease site. Inclusion of the MD studies with the biochemical studies reported here is recommended.

(vii) Pg. 15: The authors state that "...tempting to suggest that these variants too, limit the ability of Pol to accommodate..." It is a stretch to imagine smaller substitutions such as P286S/H restrict access to the exonuclease site by a gating mechanism similar to P286R. This is merely a speculation and the statement needs to be revised or deleted.

(viii) Pg 15: " ...Pol spends a significantly larger proportion of time with DNA bound in the exonuclease mode.." should be deleted since there is no evidence to support this statement.

(ix) Pg. 16: The authors note "This view is supported by a multitude of studies, including strand-specific...". This is not quite accurate since there is evidence for both Pols and as the leading strand polymerase. As the authors themselves state later in the text, and as evident from this study on Pol-P286R, it remains possible that the Pol mutants used to establish its role as the primary leading strand polymerase may not reflect the polymerase and exonuclease activities of the native / wild type counterpart. In light of this ambiguity, the authors should cite the following papers which support an alternate model whereby Pol serves as the leading strand polymerase.

Johnson RE, et al: A major role of DNA polymerase delta in replication of both the leading and lagging DNA strands. *Mol Cell* (2015) 59(2): 163-175.

Stillman B: Reconsidering DNA polymerases at the replication fork in eukaryotes. *Mol Cell* (2015) 59(2): 139-141.

Waga S, Stillman B: Anatomy of a DNA replication fork revealed by reconstitution of sv40 DNA replication in vitro. *Nature* (1994) 369(6477):207-212.

(x) Pg. 16: Regarding "Our data suggest two possibilities. First, the previously..." Again, these studies on Pol-P286R reported here and the inferred mechanism underlying its pathogenicity may not be universally applicable to all mutants. Any such generalization should be eliminated from the discussion.

Reviewer #2 (Remarks to the Author):

The study by Xing, et al. entitled "A recurrent cancer-associated substitution in the exonuclease domain of DNA polymerase epsilon produces a hyperactive enzyme" presents a comprehensive study of the pol epsilon -P286R yeast analog pol epsilon -P301R, which is a common severe mutator in tumors. The study concludes the major consequences of the P301R mutation is a dramatically increased DNA polymerase activity on natural, mismatched, and secondary structure containing substrates. This is a striking and interesting result for a biologically relevant point mutation in pol epsilon and is supported by strong biochemistry and in vivo yeast studies. Overall, the study is well

written and of interest to the field and advances our understanding of the balance between the catalytic and exonuclease activities of pol epsilon. The authors also did a very nice job with the discussion. The major comment to the authors are listed below.

1) The authors conclude the P301R pol epsilon mutation results in a hyperactive DNA polymerase based on the results presented in Figure 3 and subsequent figures. The data as presented does indicate the mutation results in an increased qualitative activity of the enzyme relative to the wild-type and exo- pol epsilon enzymes. However, it is not evident if the authors did an active site titration to standardize the respective active fractions between each protein preparation used (P301R, WT, exo-). It would be reassuring if each protein preparation had a similar amount of active enzyme. While this is not a common control, it is important for this study to ensure that each experiment variant enzyme prep has the same concentration of active enzyme, as only this portion is capable of nucleotide incorporation within the catalytic active site of pol epsilon. This is especially true given the major result of the study is based on this observed increase in activity of the enzyme. This can be done using a standard pre-steady state kinetic approach. Of note, the authors did indicate the total protein concentration is the same between each experiment and it is not necessary to do the exo active site titration under the given model presented in figure 6.

2) In the discussion the authors say, in reference to the companion Parkash et al study, that the only distinction between the two enzymes is the presence of steric hindrance in the mutant structure. This is not totally accurate, as there were also reported differences in metal binding (B-site) and coordination of the catalytic triad (R301 coordinates E292) that could also contribute to changes in catalysis and/or the position of the DNA between the two active sites. This should be discussed.

Reviewer #1:

(i) On Pg. 3, the authors note that "...the idea that the pathogenicity of Pol variants results from adverse effects...". This should be changed to "...the idea that the pathogenicity of all Pol exonuclease variants result solely from adverse effects..."

We agree and have revised this sentence.

(ii) Pg. 6, Fig. 1a and 1b: What happens to the exonuclease reactions at time periods longer than 10 min and whether the variant enzyme is able to catch up with the WT protein? Authors should consider including exonuclease activity from mismatched template-primer termini and internal mismatches.

We included analysis of later time points in Fig. 1a and 1b. Hydrolysis by Pol ϵ -P301R was still inefficient in comparison to the WT Pol epsilon even after 2 h incubation. We also included analysis of exonuclease activity on substrates with terminal and internal mismatches in Supplementary Figure 2 and a brief description of these results on p. 6.

(iii) On Pg. 10, "Since Pol-P301R has limited exonuclease..." should be changed to "Since Pol-P301R has residual exonuclease..."

We have done this.

(iv) Pg. 9. The authors report the ability of Pol-P301R to extend from mismatched primer-template termini. For the sake of completion, it is strongly recommended that the authors include DNA synthesis activity from template-primer duplexes harboring mismatches downstream of the primer terminus.

We have done these experiments. They are now described in a new Supplementary Figure 6 and in the text on p. 10. The increased mismatch extension ability of Pol ϵ -P301R variant in comparison to exo⁻ Pol ϵ was most pronounced with mismatches at -1 and -2 positions, with the difference gradually decreasing as the mismatch was moved further away from the primer terminus.

(v) Pg. 13-14: The authors suggest that "...inability to accommodate single stranded DNA in the exonuclease site forces the enzyme to stay in the polymerization mode, ...". This should be rephrased since there are two possibilities – the enzyme to stay in the polymerization mode or to release the template-primer duplex.

We revised this sentence to indicate that this is a proposed scenario explaining the higher DNA polymerase activity of Pol ϵ -P301R. Dissociation of the enzyme from the template-primer cannot contribute to increased activity, therefore, we do not consider this possibility as part of our model.

(vi) Pg. 13-14: While the biochemical studies reported here are technically sound, interpretation of these results are dependent on the molecular dynamics analysis of the

P301R variant with single stranded DNA modeled at the exonuclease site. Inclusion of the MD studies with the biochemical studies reported here is recommended.

If we understand correctly, the reviewer refers to the molecular dynamics data described in the companion manuscript by Erik Johansson's and Lynn Kamerlin's groups. We did not participate in this work and are not at liberty to include their data into our manuscript. We consider their findings in the discussion, because the synergy between the two studies creates a unique opportunity to provide the readers with a higher level of understanding than our study would do alone. However, the main focus of our paper is the discovery that the cancer-associated P301R variant converts Pol ϵ into a hyperactive enzyme, and this is what distinguishes it from the much less pathogenic exonuclease-deficient variant. We do not believe that the MD data are required for this discovery to have an impact on the field.

(vii) Pg. 15: The authors state that "...tempting to suggest that these variants too, limit the ability of Pol to accommodate..." It is a stretch to imagine smaller substitutions such as P286S/H restrict access to the exonuclease site by a gating mechanism similar to P286R. This is merely a speculation and the statement needs to be revised or deleted.

We agree that the structural effects of the smaller P286S/H substitutions are unlikely to be the same as those of P286R, but their mutator effects are also an order-of-magnitude smaller (Barbari et al. 2018, G3, 8, 1019). So, if P286S/H do obstruct the movement of the primer terminus to the exonuclease active site as we suggest, this would only need to be by a small fraction of what is happening in P286R. We do not think this suggestion is unreasonable. We would like to note, however, that our phrase does not refer specifically to P286S/H, as these are not particularly recurrent mutations. There are several others that are seen much more often, for example, S459F, P436R, etc., and our sentence primarily refers to those. We also took care to not say that these other changes restrict binding at the exonuclease site via the same mechanism as P286R. We recognize that the structural changes will be unique with each variant. Our goal was to suggest that the higher-than-proofreading-deficiency mutator effects of these variants may be similarly caused by increased polymerase activity due to reduced partitioning of the DNA between the exonuclease and polymerase sites. We believe our sentence adequately conveys this idea.

(viii) Pg 15: "...Pol spends a significantly larger proportion of time with DNA bound in the exonuclease mode..." should be deleted since there is no evidence to support this statement.

We have deleted this sentence.

(ix) Pg. 16: The authors note "This view is supported by a multitude of studies, including strand-specific...". This is not quite accurate since there is evidence for both Pols and as the leading strand polymerase. As the authors themselves state later in the text, and as evident from this study on Pol-P286R, it remains possible that the Pol mutants used to establish its role as the primary leading strand polymerase may not reflect the polymerase and exonuclease activities of the native / wild type counterpart. In light of this ambiguity, the authors should cite the following papers which support an alternate model whereby Pol

serves as the leading strand polymerase.

Johnson RE, et al: A major role of DNA polymerase delta in replication of both the leading and lagging DNA strands. Mol Cell (2015) 59(2):163-175.

Stillman B: Reconsidering DNA polymerases at the replication fork in eukaryotes. Mol Cell (2015) 59(2):139-141.

Waga S, Stillman B: Anatomy of a DNA replication fork revealed by reconstitution of sv40 DNA replication in vitro. Nature (1994) 369(6477):207-212.

We have cited the studies by Stillman and Prakash groups in the revised manuscript.

(x) Pg. 16: Regarding "Our data suggest two possibilities. First, the previously..." Again, these studies on Pol-P286R reported here and the inferred mechanism underlying its pathogenicity may not be universally applicable to all mutants. Any such generalization should be eliminated from the discussion.

"The previously studied variants" refers to exo^- Pol ϵ and Pol ϵ -M644G mentioned in the previous sentence in this context, not to Pol ϵ -P301R. We clarified this by specifying the names of these variants again, hopefully eliminating any ambiguity.

Reviewer #2:

1) The authors conclude the P301R pol epsilon mutation results in a hyperactive DNA polymerase based on the results presented in Figure 3 and subsequent figures. The data as presented does indicate the mutation results in an increased qualitative activity of the enzyme relative to the wild-type and exo^- pol epsilon enzymes. However, it is not evident if the authors did an active site titration to standardize the respective active fractions between each protein preparation used (P301R, WT, exo^-). It would be reassuring if each protein preparation had a similar amount of active enzyme. While this is not a common control, it is important for this study to ensure that each experiment variant enzyme prep has the same concentration of active enzyme, as only this portion is capable of nucleotide incorporation within the catalytic active site of pol epsilon. This is especially true given the major result of the study is based on this observed increase in activity of the enzyme.

This can be done using a standard pre-steady state kinetic approach. Of note, the authors did indicate the total protein concentration is the same between each experiment and it is not necessary to do the exo^- active site titration under the given model presented in figure 6.

We agree that this is an important control. We have done the titration experiments using a modification of the approach used by Goodman and co-authors (Cai et al, 1995, J. Biol.

Chem. 270: 15327-35). We present these data in Supplementary Figure 5. The fractions of active enzyme are comparable between wild-type Pol ϵ , exo⁻ Pol ϵ and Pol ϵ -P301R preparations.

2) In the discussion the authors say, in reference to the companion Parkash et al study, that the only distinction between the two enzymes is the presence of steric hindrance in the mutant structure. This is not totally accurate, as there were also reported differences in metal binding (B-site) and coordination of the catalytic triad (R301 coordinates E292) that could also contribute to changes in catalysis and/or the position of the DNA between the two active sites. This should be discussed.

We have discussed this in the revised manuscript (p. 13).

REVIEWERS' COMMENTS:

Reviewer #1 (Remarks to the Author):

In light of the changes made by the authors in the revised manuscript "A recurrent cancer-associated substitution in the exonuclease domain of DNA polymerase ϵ produces a hyperactive enzyme", I recommend the manuscript for publication in Nature Communications without any major changes. A minor note:

In reference to comment (vii), authors should consider specifying mutations such as S459F, P436R explicitly on Pg. 15 to avoid any confusion.

Reviewer #2 (Remarks to the Author):

The authors addressed my prior concerns. This is a novel and interesting study worthy of publication in Nature Communications.

Response to reviewer's comments:

Reviewer #1:

In reference to comment (vii), authors should consider specifying mutations such as S459F, P436R explicitly on Pg. 15 to avoid any confusion.

We carefully considered the reviewer's suggestion and chose to make no changes to our sentence. The list of these mutations is quite extensive (over a dozen); specifying them all in this sentence is not productive. We believe the reference to the review by Rayner et al. is much more helpful for the reader. The review has a structural image showing the location of these mutations in respect to the DNA binding cleft, as well as details on their incidence in tumors.